# Clinical Study of Intraoperative Microelectrode Recordings during Awake and Asleep Subthalamic Nucleus Deep Brain Stimulation for Parkinson’s Disease: A Retrospective Cohort Study

**DOI:** 10.3390/brainsci12111469

**Published:** 2022-10-29

**Authors:** Guang-Rui Zhao, Yi-Feng Cheng, Ke-Ke Feng, Min Wang, Yan-Gang Wang, Yu-Zhang Wu, Shao-Ya Yin

**Affiliations:** 1Clinical College of Neurology, Neurosurgery and Neurorehabilitation, Tianjin Medical University, Tianjin 300070, China; 2Department of Neurosurgery, Lu’an Hospital Affiliated to Anhui Medical University, Lu’an 237000, China; 3Department of Functional Neurosurgery, Huanhu Hospital, Tianjin University, Tianjin 300350, China; 4Department of Neurology, Huanhu Hospital, Tianjin University, Tianjin 300350, China

**Keywords:** microelectrode recording (MER), subthalamic nucleus (STN), deep brain stimulation (DBS), Parkinson’s disease (PD), general anesthesia (GA), local anesthesia (LA), asleep DBS, awake DBS

## Abstract

Our objective is to analyze the difference of microelectrode recording (MER) during awake and asleep subthalamic nucleus deep brain stimulation (STN-DBS) for Parkinson’s disease (PD) and the necessity of MER during “Asleep DBS” under general anesthesia (GA). The differences in MER, target accuracy, and prognosis under different anesthesia methods were analyzed. Additionally, the MER length was compared with the postoperative electrode length by electrode reconstruction and measurement. The MER length of two groups was 5.48 ± 1.39 mm in the local anesthesia (LA) group and 4.38 ± 1.43 mm in the GA group, with a statistical significance between the two groups (*p* < 0.01). The MER length of the LA group was longer than its postoperative electrode length (*p* < 0.01), however, there was no significant difference between the MER length and postoperative electrode length in the GA group (*p* = 0.61). There were also no significant differences in the postoperative electrode length, target accuracy, and postoperative primary and secondary outcome scores between the two groups (*p* > 0.05). These results demonstrate that “Asleep DBS” under GA is comparable to “Awake DBS” under LA. GA has influences on MER during surgery, but typical STN discharges can still be recorded. MER is not an unnecessary surgical procedure.

## 1. Introduction

Currently, deep brain stimulation (DBS) has become the optimal surgical procedure of choice for advanced Parkinson’s disease (PD) [1,2]. The subthalamic nucleus (STN) is the target most frequently used for DBS in PD, and STN-DBS has evolved as a mainstream therapy for PD [2]. Traditionally, a DBS procedure is performed with the patient awake based on local anesthesia (LA), followed with an intraoperative microelectrode recording (MER) and/or macro-electrode test stimulation to validate the optimal location of the electrode. However, this procedure could be burdensome for patients who cannot tolerate a prolonged drug withdrawal and remaining supine for long periods of time, as well as those with severe anxiety, obstructive sleep apnea, claustrophobia, and uncontrolled hypertension [3]. In additional, “Awake DBS” surgery increased the risk of intracranial hemorrhage due to the use of more microelectrode tracks [4], limiting some potential DBS candidates.

Recently, magnetic resonance imaging (MRI) can more clearly visualize STN, making the target location more accurate. Therefore, “Asleep DBS” under general anesthesia (GA) relying on the image location is adopted by more and more centers, and even some centers consider that MER is not a necessary procedure [4,5], while many others still insist on this application under GA [6,7]. The related research points out that “Asleep DBS” may be better for improving motor symptoms than “Awake DBS”, and “Asleep DBS” has more advantages in speech fluency and quality of life, while, MER does not add more meaningful value to DBS surgery [4]. In addition, Maria Inês Soares et al. [5] reported that MER was not consistent with the final electrode position, and MER might not be necessary to obtain good clinical outcomes in PD undergoing STN-DBS. At present, there are many disputes about the application of intraoperative MER.

In our central study, regardless of the anesthesia methods, MER was used in a surgical procedure as a method of target verification. In this study, the authors retrospectively analyzed the single center data of undergoing STN-DBS for PD and reconstructed leads to measure the postoperative electrode length, which was rarely reported in the past. By comparing and analyzing the electrophysiological characteristics and clinical prognosis of “Awake DBS” and “Asleep DBS”, our objective is to further elucidate the characteristics and necessity of MER under different anesthesia methods.

## 2. Materials and Methods

### 2.1. Participants

In this retrospective case series, we examined 50 consecutive idiopathic PD patients who underwent STN-DBS surgery at the Tianjin Huanhu Hospital between 1 April 2020 and 15 January 2022. Excluding 4 patients with missing images or MER dates, and 3 patients who failed to follow-up, a total of 43 patients were included. Twenty-three patients who were operated on under “Awake DBS” were included in the LA group, including 20 patients with a bilateral electrode implantation and 3 patients with a unilateral (left) electrode implantation. In the LA group, there were 43 electrodes. Twenty patients operated on under “Asleep DBS” were included in the GA group and they had all undergone a bilateral operation, including 40 electrodes (Figure 1). The study was approved by the ethics board of Tianjin Huanhu Hospital (No. 2021-059).

### 2.2. Target Planning

For STN localization, the indirect method was based on the anterior–posterior commissure (AC-PC) coordinates as the initial guidance, and the targeting was adjusted based on the anatomical visualization of the structures. Direct targeting was carried out as a preoperative MRI co-registered with intraoperative CT (iCT) sequences after the Leksell G frame was placed under LA. Images fusion was performed at the surgical planning system workstation (StealthStation, Medtronic or SinoPlan, Sinovation). The dorsolateral STN targeting and trajectories were directly visualized in the maximum diameter plane of the red nucleus relative to the line of the anterior–posterior commissure, as previously described [8].

### 2.3. Electrophysiology Protocol

An intraoperative MER by LeadPoint (Medtronic) was performed for both groups. In the GA group, 1% propofol 2–4 mL/kg/h and remifentanil 0.5–1.0 μg/kg/min were pumped to maintain anesthesia. Bispectral index monitoring (Bis) was performed during the operation, and the Bis was maintained ≥70 at the beginning of the MER. Microelectrodes were implanted into the guide tube and inserted into the target via a Microdrive. The MER was recorded in 1 mm steps from 10 to 5 mm above the target, 0.5 mm steps until substantia nigra (SN) activity appeared, or until the STN activity disappeared. The MER was analyzed and recorded by two specialized electrophysiologists during the operation. There are two typical forms of STN nuclear discharges. The lateral STN showed continuous and irregular high-frequency discharges with a high density of local neurons, while the medial STN showed a continuous and irregular medium and high-frequency discharges [9,10]. The discharge in the reticular part of the SN is regular, continuous, and high frequency, while the density of the local neurons is lower than that in the STN [11] (Figure 2).

### 2.4. Measurements

A routine iCT examination was performed to determine the intracranial conditions and to be co-registered with a preoperative MRI to visualize the placement of the final electrode (four contacts with a 1.5 mm length, 0.5 mm interval, Pins model L301S). One week after the operation, a postoperative CT examination was performed, and the preoperative MRI was co-registered again. The final target was still located at the largest slice of the red nucleus, and the center of the lead artifact was defined as the final target. The vector of deviation (d) was defined as the Euclidean distance between the preoperative planning target and the final target. The algorithm of the vector of deviation (d) can be referred to as Nowacki A et al. did in [12]. The postoperative electrodes were reconstructed from the Probe’s eye view of the surgical planning system, and the electrode length in the STN was measured (Figure 3).

### 2.5. Clinical Outcome Assessment

One week before the operation, the same specialist neurologist conducted various assessments of the baseline level of patients without any medication (OFF-medication). Six months postoperatively, Patients under the state of On-stimulation(DBS turned on)/OFF-medication returned to the programing clinic for an evaluation. (I) The primary outcome investigates the motor symptoms measured by the third part of the Unified Parkinson’s Disease Rating Scale (UPDRS III), with higher scores being associated with a more severe condition. (II) The secondary outcomes measure mainly includes the following: (a) a cognitive evaluation: using the Mini-Mental State Examination (MMSE) and Montreal Cognitive Assessment (MoCA).The MMSE assesses the overall cognitive function, with a total score of 30 points, and if ≤24 points, there is a cognitive dysfunction. The MoCA is used to quickly screen a mild cognitive impairment, with a total score of 30 points, and a score < 26 points was considered to be a mild cognitive impairment. (b) The improvement of the quality of life compared with the baseline was assessed by a 39-item Parkinson’s questionnaire (PDQ39), with specific score calculation rules as referred to by Peto V et al. [13]. (c) Emotional assessment: the Hamilton Depression Scale (HAMD) and Hamilton Anxiety Scale (HAMA) were used. (III) An evaluation of anti-Parkinson’s disease drugs dependence: the application of anti-Parkinson’s drugs before and after surgery was recorded in detail and converted to the Levodopa equivalent daily dosage (LEDD). The LEDD was calculated following Tomlinson CL et al. [14].

### 2.6. Statistical Analysis

The counting data were expressed as a relative constituent ratio (%) or rate (%) and was tested by two Chi-square tests. The quantitative data were described as the mean ± standard deviations (*SD*). The Kolmogorov–Smirnov test was used for the normality test. Two independent-sample *t* tests or paired-sample *t* tests were used for the data of normal distribution with the correction applied as appropriate based on Levine’s test for equality. The Mann–Whitney *U*-test was used for the non-normal distribution data. The significance was set at *p* < 0.05 for all tests. A statistical analysis was performed using SPSS software (SPSS statistics 26.0; SPSS Inc. Chicago, IL, USA).

## 3. Result

### 3.1. Patients

There were 23 patients in the LA group, including 20 patients with a bilateral implantation and 3 patients with a unilateral (all on the left) implantation, and a total of 43 electrodes. In the GA group, all 20 patients underwent a bilateral operation, with a total of 40 electrodes. There were some differences in the baseline characteristics between the two groups but without a statistical significance (*p* > 0.05) (Table 1). In the GA group, two patients underwent the electrodes implantation first, and implanting the pulse generator was performed after a satisfactory curative effect was observed by an external temporary pulse generator stimulation. In the LA group, the electrodes and a pulse generator were implanted at one time after a satisfactory macro-stimulation.

### 3.2. Measurement of Electrode Length and MER Length

According to the electrode reconstruction from the Probe’s eye view post-operation, the electrode length penetrating the STN was measured. The average length was 4.69 ± 1.14 mm in the LA group, and 4.47 ± 1.13 mm in the GA group, without a significant difference between the two groups (*p* = 0.38). The intraoperative MER length was significantly greater than the postoperative measured electrode length (*p* < 0.01) in the LA group, while there was no statistically significant difference between the intraoperative MER length and the postoperative electrode length in the GA group (*p* > 0.05). The vector of deviation(d) targeting the STN was also measured. The vector of deviation(d) was 1.19 ± 0.58 mm in the LA group and 1.18 ± 0.57 mm in the GA group, with no significant difference between the two groups (*p* > 0.05) (Table 2) (Figure 4).

### 3.3. Comparison of Prognosis and LEED between the Two Groups

A retrospective comparison of follow-up results six months after the operation showed that the UPDRS III, the primary outcome, was significantly improved in both groups under the state of “On-stimulation/Off-medication”. The LEED of the two groups also decreased significantly. There was no significant difference in the UPDRS III score and the LEED six months after the operation between the two groups (*p* > 0.05). The secondary outcomes, including the cognitive, emotion, and 39-item Parkinson’s questionnaire, were compared without a significant difference between the two groups (*p* > 0.05) (Table 3).

## 4. Discussion

The “Awake DBS” implication is to use some forms of feedback from the patient to confirm the target, as implemented by the intraoperative MER and intraoperative macro-stimulation tests. “Asleep DBS” means that the patient did not participate in the clinical evaluation during the operation, but the MER recording can still be performed during the operation [7]. In the past, more “Asleep DBS” was implemented under the guidance of imaging, and the necessity of the MER is denied when a good nucleus discharge cannot be recorded [4,5]. However, the errors caused by brain shift and the image fusion cannot be ignored. It was reported that the basal ganglia structure shifted 0.6 mm after the myringotomy and 9% of patients could shift more than 2 mm [15]. The method of intraoperative CT (iCT) fusion with a preoperative MRI is the mainstream method of DBS electrode placement, with an average fusion error of 1.25 mm [16,17]. Moreover, the use of a postoperative re-examination CT, and the fusion with a preoperative MRI to verify the target, may lead to greater errors. For this reason, the application of the MER in DBS is controversial.

In this cohort, the MER in the GA group was significantly weaker than that in the LA group, and the discharge frequency and amplitude were decreased. The measured MER discharge length in the GA group was significantly lower than that in the LA group (*p* < 0.01). However, there was no significant difference in the electrode length within the STN between the two groups by the postoperative Probe’s eye view electrode reconstruction (*p* = 0.38). Furthermore, there was no significant difference between the postoperative electrode length and the MER length in the GA group (*p* = 0.67). However, in the LA group, the MER length was larger than the postoperative electrode length (*p* < 0.01). The authors speculated that good nucleus discharges under LA may make it difficult to distinguish the boundary between the STN and SN, which may lead to the excessive estimation of the STN electrophysiological signal length. Previous studies have found that the measured electrode length is not consistent with the intraoperative MER length by measuring the electrode length in the preoperative surgical plan and comparing it with the intraoperative MER under LA and pointed out that the measured electrode length did not represent the length of the MER [12]. However, under GA, a report pointed out that there was no statistical difference between the pre-operative electrode length and the intraoperative MER length and considered that the MER was of a great significance in selecting the puncture track [6]. These results are consistent with this study, but we used the postoperative electrode reconstruction from the Probe’s eye view to measure the electrode length in the STN, which is more convincing than the previous studies. In addition, we compared the targeting vector of deviation between the GA and LA groups, without significant differences (*p* = 0.95). This indicates that although the discharge of the STN has been affected during the “Asleep DBS” under GA, the typical STN discharge can still represent a good electrode placement, rather than pursuing the same discharge frequency and amplitude, or even the signal length, as “Awake DBS” under LA. Some studies have pointed out that the MER must be greater than 4 mm to predict a good target and prognosis, but this study recorded the MER length under LA, which may not be applicable to GA [18].

In this cohort, we also compared the prognosis of two groups. We found that the primary outcome, UPDRS III, and the secondary outcomes, including PDQ39, and the cognition and emotion of the two groups after surgery, were not statistically significant. This is consistent with the conclusions of previous researchers [19,20,21,22]. A report pointed out that the MER signal length of the STN could not predict a prognosis, and it considered that even longer MER signals did not represent the optimal target position, located at the dorsolateral STN [23,24]. Furthermore, Wodarg et al. reported that the electrode position of imaging can better predict the long-term clinical results than the MER data [25]. In our study, it was also found that although the MER length of the LA group was significantly longer than that of the GA group, there were no significant differences in the primary and secondary prognosis between the two groups. A longer MER does not mean a better prognosis, which is consistent with the above conclusions. In addition, the boundary between the electrical activity of the STN and SN is often difficult to distinguish under LA, which may eventually lead to an electrode placement which is too deep. Some researchers indicated that the SN may be stimulated due to a too deep electrode placement, which may lead to various adverse reactions, including mania and emotional disorders, such as depression [26,27]. However, some researchers believed that the stimulation of the SN does not significantly affect the prognosis, and even put forward the idea of a dual targets’ stimulation of the STN and SN [28,29]. In our study, there was no difference in the emotional response between the two groups after surgery, which may be due to the short follow-up time, so we still need further follow-up observation for these conclusions.

### Limitation

There are still some limitations in our study. First, our research is retrospective, but because our research scheme involves a postoperative image reconstruction and comparison with an intraoperative MER, so the nature of study was determined, it was difficult to adopt a prospective study. Second, the grouping of LA and GA was mainly based on the patients’ willingness, rather than a random allocation. Moreover, the follow-up time is six months, which is still short, and the observation time for the emotional, cognitive, and other changes may be insufficient. Finally, our anesthesia maintenance protocol was classic propofol + remifentanil, and the Bis > 70 was maintained during the MER. At present, it has been reported that inhaled anesthetic drugs such as desflurane and sevoflurane and dexmedetomidine, which are used for anesthesia maintenance, have different impacts on electrophysiology [30,31], so our research conclusions may not be suitable for MER and electrode implantation under other anesthesia regimens.

## 5. Conclusions

The “Asleep DBS” procedure under GA was similar to the classic “Awake DBS” procedure under LA in terms of the target accuracy, electrode length within the STN, and prognosis. However, even in the case of the Bis > 70, GA still has a certain impact on an intraoperative MER, but the typical STN discharge can still predict the precise target position. The “Asleep DBS” procedure under the GA increases the comfort and cooperation degree of patients and expands the beneficiary population. Moreover, MER is helpful to accurately locate and verify the target during surgery.

## Figures and Tables

**Figure 1 brainsci-12-01469-f001:**
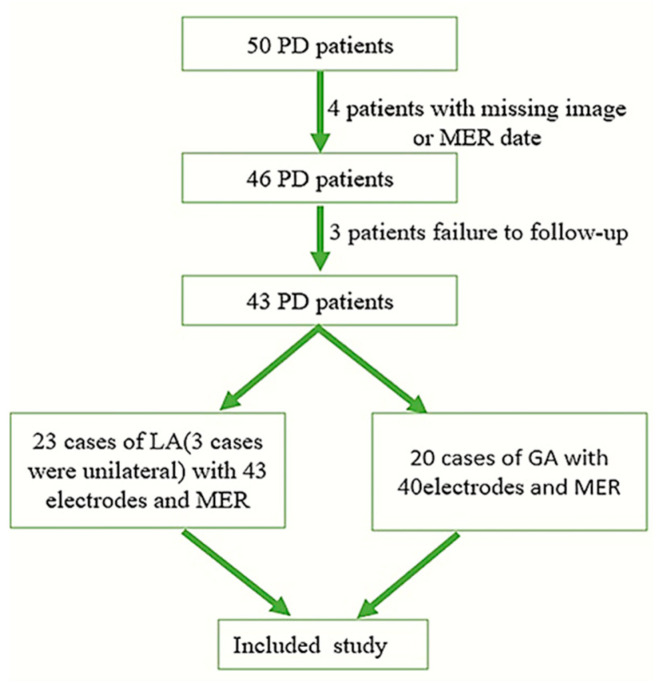
The patients screening procedure. PD, Parkinson’s disease; MER, microelectrode recording; and STN, subthalamic nucleus.

**Figure 2 brainsci-12-01469-f002:**
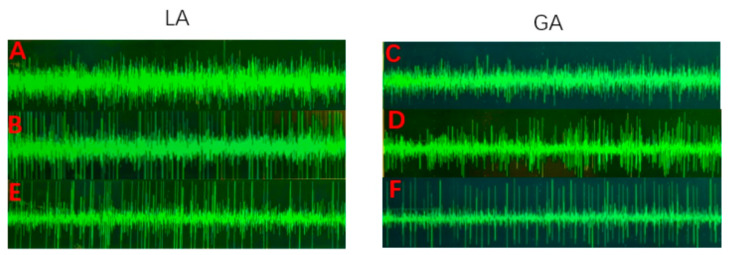
The MER of typical STN and SN under different anesthesia methods. (**A**). The medial of STN discharge under LA. (**B**). The lateral of STN discharge under LA. (**C**). The medial of STN discharge under GA. (**D**). The lateral of STN discharge under GA. (**E**). SN discharge under LA. (**F**). SN discharge under GA. MER, microelectrode recording; STN, subthalamic nucleus; SN, substantia nigra; GA, general anesthesia; and LA, local anesthesia.

**Figure 3 brainsci-12-01469-f003:**
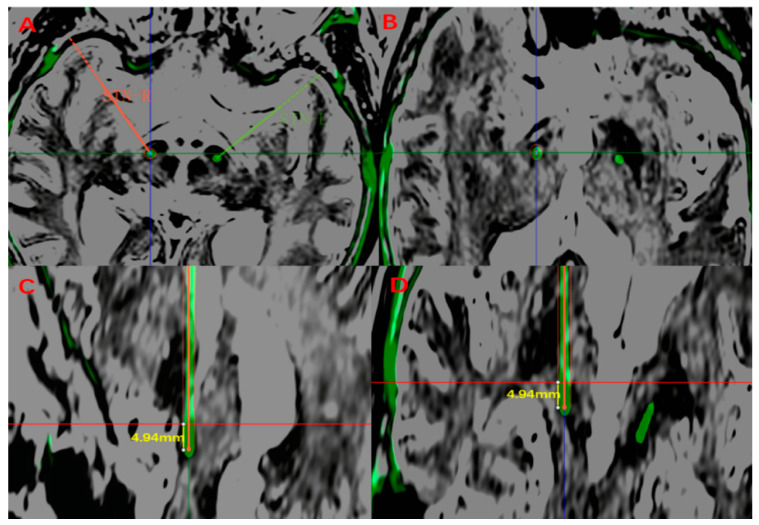
The electrode was reconstructed and measured postoperatively. Reconstructed views along the planned trajectory (Probe’s eye view) of 1 representative patient are shown to demonstrate the measurement of electrode. (**A**). The postoperative target validation of right STN. (**B**). Showing the entry point of STN by Probe’s eye view. (**C**,**D**). The electrode length was measured from the entry point slice (shown by the red horizontal line) to the end of planned track, (sagittal and coronal view).

**Figure 4 brainsci-12-01469-f004:**
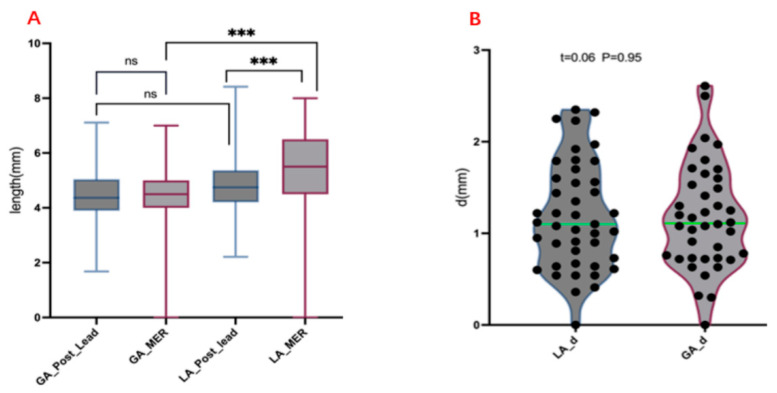
(**A**) Cross comparison of electrode length and MER length between two groups. (**B**) Comparison of the vector of deviation (d) between two groups. MER, microelectrode recording; d, vector of deviation; GA, general anesthesia; LA, local anesthesia; and ns, not significant; *** *p* < 0.01.

**Table 1 brainsci-12-01469-t001:** Baseline patient characteristics of the two group.

	GA (*n* = 20)	LA (*n* = 23)	*p* Value
Sex ratio (M/F)	12/8	14/9	0.95
Age in years (mean ± *SD*)	68.05 ± 8.62	64.48 ± 5.95	0.07
Years since PD diagnosis	7.60 ± 2.06	7.22 ± 3.53	0.22
Baseline UPDRS III (mean ± *SD*)	50.31 ± 19.64	47.70 ± 15.18	0.63
Baseline LEDD (mean ± *SD*)	711.16 ± 375.34	790.88 ± 346.31	0.47
Duration of use of medication (mean ± *SD*)	5.71 ± 3.07	6.05 ± 3.27	0.72
PDQ39(mean ± *SD*)	32.86 ± 17.77	29.62 ± 20.82	0.59
HAMA (mean ± *SD*)	12.95 ± 8.08	11.52 ± 8.53	0.43
HAMD (mean ± *SD*)	14.05 ± 8.16	11.91 ± 8.37	0.31
MoCA (mean ± *SD*)	23.95 ± 3.40	24.52 ± 2.98	0.78
MMSE (mean ± *SD*)	26.10 ± 2.45	26.09 ± 2.78	0.87

GA, general anesthesia; LA, local anesthesia; M, male; F, female; *SD*, standard deviations; UPDRS III, Unified Parkinson’s Disease Rating Scale III; LEDD, Levodopa equivalent daily dosage; PDQ39, 39-item Parkinson’s questionnaire; HAMA, Hamilton Anxiety Scale; HAMD, Hamilton Depression Scale; MoCA, Montreal Cognitive Assessment; and MMSE, Mini-Mental State Examination.

**Table 2 brainsci-12-01469-t002:** Comparison of the vector of deviation (d) between two groups; cross comparison of electrode length and MER length between two groups.

	Post-Lead Length (mm)	MER Length (mm)	d (mm)	Post-Lead Length vs. MER Length
LA (*n* = 43)	4.69 ± 1.14	5.48 ± 1.39	1.19 ± 0.58	T = −4.27 *p* = 0.001
GA (*n* = 40)	4.47 ± 1.13	4.38 ± 1.43	1.18 ± 0.57	T = 0.50 *p* = 0.61
*p*	0.38	0.001	0.95	

MER, microelectrode recording; d, vector of deviation.

**Table 3 brainsci-12-01469-t003:** Comparison of the prognosis between the two groups after six months.

	GA	LA	*p* Value
Post-LEED (mean ± *SD*)	507.64 ± 207.01	539.61 ± 129.81	0.56
Post-UPDRS III (mean ± *SD*)	24.40 ± 10.69	25.00 ± 9.96	0.85
Post-HAMA (mean ± *SD*)	10.55 ± 4.99	9.61 ± 6.15	0.36
Post-HAMD (mean ± *SD*)	11.05 ± 4.57	10.78 ± 6.94	0.43
Post-MoCA (mean ± *SD*)	23.75 ± 3.70	24.74 ± 2.53	0.49
Post-MMSE (mean ± *SD*)	25.80 ± 2.33	25.70 ± 2.84	0.99
Post-PDQ39 (mean ± *SD*)	30.59 ± 16.23	23.46 ± 15.78	0.15

GA, general anesthesia; LA, local anesthesia; *SD*, standard deviations; UPDRS III, Unified Parkinson’s Disease Rating Scale III; LEDD, Levodopa equivalent daily dosage; PDQ39,39-item Parkinson’s questionnaire; HAMA, Hamilton Anxiety Scale; HAMD, Hamilton Depression Scale; MoCA, Montreal Cognitive Assessment; and MMSE, Mini-Mental State Examination.

## Data Availability

The data presented in this study are available on request from the corresponding author. The data are not publicly available due to privacy.

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
