# Peer review of "Clinical Study of Intraoperative Microelectrode Recordings during Awake and Asleep Subthalamic Nucleus Deep Brain Stimulation for Parkinson’s Disease: A Retrospective Cohort Study"

_brainsci, 2022, doi:10.3390/brainsci12111469_

Round 1
Reviewer 1 Report
The present study is in line with recent investigations about the role of asleep DBS and how its outcome compares to awake surgery. Comparable outcomes are confirmed and provide further evidence of asleep DBS surgery as equally effective than awake surgery. The novelty of this work is that it adds information about the MER signal quality during asleep DBS and its general applicability in this setting. Again, no differences are found compared to the awake surgery cohort, which further demonstrates asleep DBS with MER-guidance as a clinically valid method.
Author Response
We sincerely thank the reviewer for your valuable feedback that we have used to improve the quality of our manuscript. The reviewer comments are” Moderate English changes required”. The authors checked the statement of the paper again and made corresponding modifications.
Reviewer 2 Report
The manuscript titled “Clinical study of intraoperative microelectrode recordings during Awake and Asleep subthalamic nucleus deep brain stimulation for Parkinson’s Disease: A Retrospective Cohort Study” By Zhou et al. compares the motor and non-motor outcome of ‘awake’ and ‘asleep’ STN-DBS surgery. Classically, STN-DBS surgery is performed in awake state, but some evidence show that the clinical outcome is comparable between these two methods. Additionally, the rate of complication was also lower in awake surgery. Hence, many groups now prefer DBS surgery under GA (asleep) as it is comfortable for the patients. Conventionally, microelectrode recording (MER) was essential to find the location of the STN. The popularity of MER has alleviated after the introduction of advanced imaging techniques and methods of Intraoperative reconstruction of images for localization. Supposedly the localization of STN through MER is less reliable than the imaging technique. However, the debate regarding the selection of anesthesia and usage of MER for localization is still alive. In this backdrop, the current study could be an important addition in the existing literature related to techniques of DBS surgery.
To my understanding the authors had three objectives of this retrospective analysis: 1. to compare the safety and motor and non-motor outcome in awake vs asleep DBS surgery 2. Characterize MER in awake and asleep surgery 3. To establish necessity of MER for localization. After carefully reading the manuscript, I found the authors could do justice to first two objectives, but I am not entirely confident about the achievement of third objective through the result. Since, it appears that the decision of localization of the STN was not dependent on MER, it is difficult to establish its relevance. Let us put it in this way - what difference of outcome would have happened in absence of intraoperative MER. To my understanding, the current research was not able to solve this query as both the groups used MER as an adjunct to intraoperative image reconstruction method for localization. May I request the authors to kindly justify their point in favor of their claim regarding non-redundancy of intraoperative MER.
Apart from this major concern - few minor findings are as follows -
1. The sentence is incomplete in line 114 (algorithm for vector of deviation….)
2. Line 145- the categorical data is presented as percentage not as mean
3. Line 212- this part of the discussion is not supported by results. The result of MER can be presented as the legend of the figure 2.
4. In line 222- the authors expressed their doubt regarding the accuracy of intraoperative MER length measurement under LA. If that is true, then how can MER be helpful in localization?
5. Line 233- please check if it will be awake or asleep.
